# Prevalence of Depressive Symptoms in the Elderly Population Diagnosed with Type 2 Diabetes Mellitus

**DOI:** 10.3390/ijerph17103553

**Published:** 2020-05-19

**Authors:** Beata Dziedzic, Zofia Sienkiewicz, Anna Leńczuk-Gruba, Ewa Kobos, Wiesław Fidecki, Mariusz Wysokiski

**Affiliations:** 1Department of Development of Nursing and Social & Medical Sciences, Medical University of Warsaw, 01-445 Warsaw, Poland; zofia.sienkiewicz@wum.edu.pl (Z.S.); anna.lenczuk-gruba@wum.edu.pl (A.L.-G.); ewa.kobos@wum.edu.pl (E.K.); 2Department of Basic Nursing and Medical Teaching, Chair of Development in Nursing, Faculty of Health Sciences, Medical University of Lublin, 20-081 Lublin, Poland; wieslaw.fidecki@umlub.pl (W.F.); mariusz.wysokinski@umlub.pl (M.W.)

**Keywords:** depression, type 2 diabetes, elderly people

## Abstract

*Introduction:* A sharp rise in the population of elderly people, who are more prone to somatic and mental diseases, combined with the high prevalence of type 2 diabetes mellitus and diabetes-associated complications in this age group, have an impact on the prevalence of depressive symptoms. *Aim of the work:* The work of the study was the evaluation of the prevalence of depressive symptoms in the elderly population diagnosed with type 2 diabetes mellitus. *Materials and methods:* The pilot study was conducted in 2019 among 200 people diagnosed with type 2 diabetes mellitus, aged 65 years and above, receiving treatment in a specialist diabetes outpatient clinic. The study was based on a questionnaire aimed at collecting basic sociodemographic and clinical data and the complete geriatric depression scale (GDS, by Yesavage) consisting of 30 questions. *Results:* The study involved 200 patients receiving treatment in a diabetes outpatient clinic. The mean age of the study subjects was 71.4 ± 5.0 years. The vast majority of the subjects (122; 61%) were women, with men accounting for 39% of the study population (78 subjects). A statistically significant difference in the GDS (*p* < 0.01) was shown for marital status, body mass index (BMI), duration of diabetes, and the number of comorbidities. Patients with results indicative of symptoms of mild and severe depression were found to have higher BMI, longer disease duration, and a greater number of comorbidities. There were no statistically significant differences in the level of HbA1c. *Conclusions:* In order to verify the presence of depressive symptoms in the group of geriatric patients with diabetes mellitus, an appropriate screening programme must be introduced to identify those at risk and refer them to specialists, so that treatment can be promptly initiated. Screening tests conducted by nurses might help with patient identification.

## 1. Introduction

Based on WHO data, diabetes mellitus is high on the list of 10 leading causes of death worldwide, with a total of 1.6 million diabetes-related deaths recorded in 2016. Depression is a serious global health threat, as the number of cases of this disease continues to rise. Globally, depression is estimated to affect 264 million people [1]. Furthermore, the elderly population continues to grow. According to Eurostat data, in 2018, people aged 65 years and older accounted for 19.7% of the total EU population, so, for every person in this age group, there are slightly more than three people of working age [2]. Since the growing number of elderly people correlates with an increase in the number of depression cases, it is important to recognise the symptoms of depression in this age group as soon as possible and initiate an appropriate treatment [3,4,5]. Depression, characterised by persistent feelings of sadness and a lack of interest or pleasure, is a leading cause of reduced functional ability and impaired quality of life, inevitably resulting in disability [1]. Diabetes mellitus, which is a chronic disease, gives rise to a number of complications, including depression, which is a common condition associated with type 2 diabetes. In addition, emotional problems commonly affecting the geriatric population exacerbate the disease [6,7,8,9,10,11]. The coexistence of disorders such as diabetes and depression in the elderly population is a major social problem. Multiple studies have shown a two-way relationship between the prevalence of diabetes mellitus and depressive symptoms [6,12,13,14]. A study conducted among elderly American Indians with type 2 diabetes mellitus found a positive correlation between depression and mortality [15]. This poses a major problem, especially since depression occurring in elderly patients is often undetected or inadequately treated [6,15,16,17].

The complexity of the problem caused by diabetes, old age, and depression remains an important public health challenge. In 2014, the prevalence of diabetes in the European population aged 65–74 years and 75 years or older was 16.3% and 19.6%, respectively [2]. Based on the 2014 data released by Statistics Poland, the prevalence of diabetes in the elderly age group in Poland was 17.7% among men and 17.2% among women. However, the number of elderly people is expected to rise further. The proportion of persons older than 60 years in the global population is projected to reach 40.4% by 2050 [18]. In recent years, there has also been a significant increase in the prevalence of depressive symptoms in this age group [19]. The percentage of people aged 65 and over who during the last month had no symptoms of malaise, felt happy and energised, and were rarely depressed and sad was 38% in 2015 [18]. An effective instrument for identifying elderly people at risk of developing depression, who are additionally affected by comorbidities, is screening based on appropriate self-assessment depression scales [6,20,21]. In the circumstances, it seems reasonable to involve entire therapeutic teams, not only psychiatrists, in the process of early assessment of the risk of depression, with prompt referrals of high-risk patients to specialists [6,19]. Benefits of this approach have been shown in one study based on cooperation between a nurse and a primary care physician [12].

The aim of the study was the evaluation of the prevalence of depressive symptoms in the elderly population diagnosed with type 2 diabetes mellitus.

## 2. Materials and Methods

The pilot study was conducted on a group of 200 patients treated at a primary healthcare diabetes outpatient clinic located in the vicinity of Warsaw, Poland in 2019. The study inclusion criteria were: aged 65 years and over, diagnosis of type 2 diabetes mellitus, and patient consent to participate in the study. The study method was a diagnostic survey based on an original questionnaire designed to collect basic sociodemographic and clinical information and a standardised research tool in the form of the geriatric depression scale (GDS; Yesavage). GDS is one of the most widely used screening tools for the self-assessment of depression in the elderly. The full 30-item version of GDS was used, with questions requiring a “yes” or “no” answer. The questions referred to the patients’ mood over the preceding two weeks. Interpretation of results: scores 0 to 9—no depression, scores 10 to 19—mild depression, and scores of 20 and above—severe depression [20]. Results obtained using the scale are not equivalent to the diagnosis of depression, but they are diagnostically significant, and patients with GDS scores indicating depression should be referred for consultation with a specialist [20]. In addition, all patients were subjected to anthropometric measurements, and their medical records were reviewed. The study was approved by the Bioethics Committee at the Medical University of Warsaw (approval no. AKBE/335/2019) and by the president of the primary healthcare outpatient clinic where the study was carried out.

In order to identify risk factors for the development of depression in patients, univariate and multivariate logistic analyses were conducted. Using the procedure of stepwise forward selection of variables at a significance level of 0.1, a set of risk factors was selected and incorporated into the final logistic regression model. The following variables were selected for the uni- and multivariate analyses: gender, comorbidities, body mass index (BMI), and HBA1C level. Nominal variables were compared using the chi-square test. For comparisons of continuous variables in 3 groups due to the lack of normal distribution (Shapiro-Wilk test), the Kruskal-Wallis test was used. Post-hoc comparisons were made using the Tukey test. The level of statistical significance was set at *p* < 0.05. The analyses were conducted using Stata 15.1 software (StataCorp, 2017, Stata Statistical Software: Release 15; StataCorp LLC., College Station, TX, USA).

The study was approved by the Bioethics Committee at the Medical University of Warsaw (approval no. AKBE/335/2019) and by the president of the primary healthcare outpatient clinic where the study was carried out.

The study was conducted in compliance with applicable ethical standards and principles.

## 3. Results

The study involved a total of 200 patients. The mean age of the study subjects was 71.4 ± 5.0 (72.1 ± 4.5 in patients with mild depression, 72.4 ± 5.0 in patients with severe depression, and 70.9 ± 5.0 in patients without depression). Most of the subjects (122; 61%) were women, with men accounting for 39% of the study population (78 subjects). The group of subjects with mild and severe depression was also dominated by women, 33 (73.3%) and 16 (76.1%), respectively, with 12 (26.6%) and 5 (23.8%) men in both groups, respectively. Most of the study subjects 93 (46.5%) had completed vocational education and were in relationships (151; 75.5%). Statistically significant differences were seen between the marital status and the presence of depressive symptoms (*p* < 0.001). Detailed information is given in Table 1.

The results of the geriatric depression scale (GDS) analysed in relation to the collected clinical data revealed statistically significant differences (*p* < 0.01) for the BMI (body mass index), duration of diabetes, and the number of comorbidities. Patients with symptoms of mild and severe depressive symptoms were found to have significantly higher BMI, longer duration of the disease, and more comorbidities. Grade 1 obesity was present in 12 (26.6%) and 10 (47.6%) study subjects with mild and severe depression, respectively. Grade 2 obesity was determined in 13 (28.8%) and 6 (28.5%) study subjects with mild and severe depression, respectively. The mean duration of diabetes in nondepressive patients was M = 10.2; in patients with mild depression, it was M = 12.7, and in patients with severe depression, M =13.6. With regard to comorbidities, arterial hypertension was observed in 38 (84.4%) subjects with mild depression and 21 (100%) subjects with severe depression. Among those without depression, arterial hypertension was identified in 63 (47.0%) cases. A total of 10 (47.6%) subjects with severe depressive symptoms were found to have inadequate glycaemic control expressed as HbA1c levels. Detailed results are shown in Table 2, Table 3 and Table 4.

In order to evaluate the relationship between depression and risk factors for the development of depression, multivariate and univariate logistic regressions were used, dividing the subjects into two groups (nondepression patients and mild and severe depression patients). Statistically significant differences were observed for the number of comorbidities: OR: 18.38, 95% CI: 5.2–67.21 and OR: 19.78, 95% CL: 5.92–66.13. The results are listed in Table 5.

## 4. Discussion

Based on Eurostat 2014 data for Poland, almost every other person older than 60 years is classified as biologically disabled. Unfortunately, the number of disabled people grows with age, and slightly more than half of the population older than 70 years comprises disabled individuals [18]. Depression is one of the leading causes of a reduction in disability-adjusted life years globally [22]. Elderly people rarely report mental health problems, so it is difficult to identify emotional difficulties which induce the progression of pre-existing diseases [21]. Despite extensive knowledge about the significant association between depression and diabetes, the problem persists and even grows worse because of the rapid growth of the elderly population. Deterioration of the quality of life, impairment of daily activities, and an increased frequency of using medical services generate healthcare costs [23].

The aim of the study was the evaluation of the prevalence of depressive symptoms in the elderly population diagnosed with type 2 diabetes mellitus. The underlying assumption was to highlight the scale of the problem and explore possible ways to resolve it by involving therapeutic teams, including nurses, in the identification of patients with depressive symptoms. Our study identified 33 (73.3%) women with mild depressive symptoms and 16 (76.1%) women with severe depressive symptoms. In men, the proportions were 12 (26.6%) and five (23.8%), respectively. An analysis of the study results revealed statistically significant differences in terms of the marital status, BMI, the number of comorbidities, and the duration of diabetes, which positively correlated with the occurrence of depressive symptoms in the study population. No correlation was found between the development of depressive symptoms and the completed level of schooling, HbA1c level, type of diabetes treatment, and gender.

Górska-Ciebiada et al. obtained similar results to those presented in this paper. In their study, a relationship was found between depressive symptoms, which were observed in 20 (36.36%) subjects, and factors including disease duration, BMI, and the number of comorbidities. However, no correlation with the HbA1c level was noted. Statistically significant differences were observed for the gender of the study subjects, which was not demonstrated in our study, though there is a clear difference between the number of women and men participating in the study and experiencing symptoms indicative of depression, with a female predominance [24]. In another study by the same authors, conducted on a group of 276 elderly patients with type 2 diabetes mellitus—of whom, 29.7% had depressive symptoms—significant factors contributing to depression were found to include gender, BMI, and a large number of comorbidities [25].

A cross-sectional study by Azniza et al. showed a considerably high prevalence of depression (32.1%) among the elderly population. Factors contributing to depression included diabetic complications, living arrangements, and the level of HbA1c, which was insignificant in our study [26].

Similar findings were reported by Féki et al. The study was conducted among 50 elderly patients (age ≥ 65 years). Depressive symptoms were identified in 34% of the subjects. Furthermore, there was a high prevalence of comorbidities (94%), including arterial hypertension (84%). In our study, arterial hypertension was diagnosed in 38 (84.4%) patients with mild depression and 21 patients with severe depression (100%). Interestingly, no patient among those with depressive symptoms (34%) received any psychiatric treatment prior to the study [27].

Perkkiö et al. conducted a study in a population consisting of 60% of the elderly population (aged 65 or older) living in the northern regions of Finland (including some that had moved away from the area). The authors found a statistically significant association between the development of depressive symptoms and previously known type 2 diabetes mellitus (T2D). The prevalence of previously known T2D was two times higher in those with than in those without depression. The frequency of depressive symptoms was assessed using the Beck Depression Inventory-II. Mild depressive symptoms (Beck Depression Inventory II (BDI-II) ≥14) were found in 7.1% of the subjects (men 9.7%; women 5.4%), while the prevalence of moderate depression (BDI >20) among men was 4.6% and, among women, 4.7%. For comparison, among the persons who had moved away from the northern regions of Finland, the prevalence of depressive symptoms was 13.7% (men 9.9%; women 17%). The majority of study participants were overweight (42.2%), and 21.7% were obese. The above results are lower than those reported in our study, which may indicate a relationship between depression and geographical areas of residence [28].

Kim et al. found an association between the prevalence and degree of depressive symptoms and factors, including higher HbA1c levels and a longer duration of diabetes mellitus. The study enrolled 155 patients (56 men and 99 women) aged ≥ 65 years. The short form of the geriatric depression scale-Korean version (SGDS-K) was used to assess the severity of depression [29]. In our study, the HbA1c levels among the subjects with severe depressive symptoms were ≥ 10% in four (19.0%) patients and ≥ 8% and <10% in 10 (47.6%) patients. Only in three (14.2%) patients, the HbA1c levels were <7%.

In the cross-sectional study by Mansori et al., involving 514 diabetic patients (381—74.1% women and 133—25.5% men) out of 532 eligible patients diagnosed with type 2 diabetes mellitus, in a slightly younger age group (most subjects older than 50 years of age), the prevalence of depressive symptoms was 46.3% (49.6% for women and 36.8% for men). Furthermore, 48.6% of diabetic patients did not have adequate glycaemic control (HbA1c > 8). Similar to our study, there was no statistically significant association between glycaemic control and depression (OR: 1.11, 95% CI: 0.87–1.57), but a positive correlation was noted for the duration of diabetes [30].

The studies discussed above, both conducted in Poland and worldwide, highlight the strong link between diabetes and depression, though it is not the only reason for depressive symptoms. Pilot studies have confirmed the underlying cause of the problem.

Looking for solutions and ways to provide the best quality of care to diabetic patients with depressive symptoms as effectively as possible, a number of useful suggestions can be found in a review by Hermanns et al. One of the models discussed in their study was based on the cooperation of the medical team, including nurses and primary care physicians, with specialists, resulting in improved clinical parameters among the study subjects. Useful information can also be derived from the assessments of the efficacy of scales used for depression screening [12].

## 5. Conclusions

The presence of depressive symptoms among geriatric patients is a factor contributing to a deterioration of their quality of life.

Planning and implementation of screening for mental health issues in the elderly population diagnosed with a lifestyle disease—such as type 2 diabetes mellitus—with existing comorbidities should be recognised as of the most important goals of the public health system.

It seems necessary to involve medical teams in the screening process in order to verify the symptoms, promptly establish the diagnosis, and initiate the appropriate depression treatment.

Screening tests conducted by nurses might help with patient identification.

## Figures and Tables

**Table 1 ijerph-17-03553-t001:** Correlations between sociodemographic variables and geriatric depression scale (GDS) scores.

Variable	Depression; Number (Proportion)			Total Number of Patients	*p*-Value (Chi-Square Test)
	None	Mild	Severe		
education					
primary	9 (6.7%)	7 (15.5%)	3 (14.2%)	19 (9.5%)	0.521
vocational	61 (45.5%)	22 (48.8%)	10 (47.6%)	93 (46.5%)	
secondary	50 (37.3%)	12 (26.6%)	7 (33.3%)	69 (34.5%)	
tertiary	14 (10.4%)	4 (8.8%)	1 (4.7%)	19 (9.5%)	
marital status					
single	3 (2.2%)	3 (6.6%)	3 (14.2%)	9 (4.5%)	<0.001
married	112 (83.5%)	31 (68.8%)	8 (38.0%)	151 (75.5%)	
divorced	3 (2.2%)	5 (11.1%)	2 (9.5%)	10 (5.0%)	
widowed	16 (11.9%)	6 (13.3%)	8 (38.0%)	30 (15.0%)	
gender					
women	73 (54.4%)	33 (73.3%)	16 (76.1%)	122 (61.0%)	0.026
men	61 (45.5%)	12 (26.6%)	5 (23.8%)	78 (39.0%)	

**Table 2 ijerph-17-03553-t002:** Correlations between clinical variables and GDS scores. BMI: body mass index.

Variable	Depression; Number (Proportion)			Total Number of Patients	*p*-Value (Chi-Square Test)
	None	Mild	Severe		
BMI					
<25	26 (19.4%)	4 (8.8)	0 (0.0%)	30 (15.0%)	<0.001
≥25 & <30	65 (48.5%)	11 (24.4%)	5 (23.8%)	81 (40.5%)	
≥30 & <35	30 (22.3%)	12 (26.6%)	10 (47.6%)	52 (26.0%)	
≥35 & <40	11 (8.2%)	13 (28.8%)	6 (28.5%)	30 (15.0%)	
≥40	2 (1.4%)	5 (11.1%)	0 (0.0%)	7 (3.5%)	
HBA1C LEVEL					
<7%	46 (34.3%)	13 (28.8%)	3 (14.2%)	62 (31.0%)	0.246
≥7% & <8%	37 (27.6%)	9 (20.0%)	4 (19.0%)	50 (25.0%)	
≥8% & <10%	38 (28.3%)	18 (40.0%)	10 (47.6%)	66 (33.0%)	
≥10%	13 (9.7%)	5 (11.1%)	4 (19.0%)	22 (11.0%)	
duration of diabetes (years)					
up to 5	11 (8.2%)	1 (2.2%)	0 (0.0%)	12 (6.0%)	0.007
6–10	72 (53.7%)	15 (33.3%)	5 (23.8%)	92 (46.0%)	
11–15	32 (23.8%)	18 (40.0%)	10 (47.6%)	60 (30.0%)	
16–20	18 (13.4%)	8 (17.7%)	5 (23.8%)	31 (15.5%)	
over 20	1 (0.7%)	3 (6.6%)	1 (4.7%)	5 (2.5%)	
diabetes treatment					
insulin	54 (40.3%)	13 (28.9%)	7 (33.3%)	74 (37.0%)	0.021
insulin + tablets	50 (37.3%)	25 (55.6%)	14 (66.7%)	89 (44.5%)	
tablets	30 (22.4%)	7 (15.6%)	0 (0.0%)	37 (18.5%)	

**Table 3 ijerph-17-03553-t003:** Correlation between depression symptoms and the duration of diabetes (years).

Variable	Depression (Mean, Standard Deviation)				*p*-Value (Kruskal-Wallis Test)
	None	Mild	Severe	Total Number of Patients	
duration of diabetes (years)	10.2 (4.3)	12.7 (4.0)	13.6(4.0)	11.1 (4.3)	<0.001
duration of diabetes	difference of means	standard deviation	95% CI		
no depression vs. mild	2.50	0.70	0.85	4.16	
no depression vs. severe	3.41	0.95	1.16	5.66	
mild vs. severe	0.91	1.07	-1.63	3.44	

**Table 4 ijerph-17-03553-t004:** Correlations between comorbidities and GDS scores.

Variable	Depression; Number (Proportion)			Total Number of Patients	*p*-Value (Chi-Square Test)
	None	Mild	Severe		
number of comorbidities					
0	65 (48.5%)	3 (6.6%)	0 (0.0%)	68 (34.0%)	<0.001
at least 1	69 (51.4%)	42 (93.3%)	21 (100.0%)	132 (66.0%)	
arterial hypertension					
no	71 (52.9%)	7 (15.5%)	0 (0.0%)	78 (39.0%)	<0.001
yes	63 (47.0%)	38 (84.4%)	21 (100.0%)	122 (61.0%)	
ischaemic heart disease					
no	94 (70.1%)	12 (26.6%)	1 (4.7%)	107 (53.5%)	<0.001
yes	40 (29.8%)	33 (73.3%)	20 (95.2%)	93 (46.5%)	
history of myocardial infarction					
no	113 (84.3%)	32 (71.1%)	7 (33.3%)	152 (76.0%)	<0.001
yes	21 (15.6%)	13 (28.8%)	14 (66.6%)	48 (24.0%)	
history of cerebral stroke					
no	123 (91.7%)	39 (86.6%)	17 (80.9%)	179 (89.5%)	0.251
yes	11 (8.2%)	6 (13.3%)	4 (19.0%)	21 (10.5%)	
heart failure					
no	128 (95.5%)	39 (86.6%)	9 (42.8%)	176 (88.0%)	<0.001
yes	6 (4.4%)	6 (13.3%)	12 (57.1%)	24 (12.0%)	
chronic kidney disease					
no	128 (95.5%)	36 (80.0%)	7 (33.3%)	171 (85.5%)	<0.001
yes	6 (4.4%)	9 (20.0%)	14 (66.6%)	29 (14.5%)	
diabetic retinopathy					
no	119 (88.8%)	26 (57.7%)	4 (19.0%)	149 (74.5%)	<0.001
yes	15 (11.1%)	19 (42.2%)	17 (80.9%)	51 (25.5%)	
diabetic foot syndrome					
no	128 (95.5%)	38 (84.4%)	18 (85.7%)	184 (92.0%)	0.032
yes	6 (4.4%)	7 (15.5%)	3 (14.2%)	16 (8.0%)	

**Table 5 ijerph-17-03553-t005:** Logistic analysis of the study group.

Variable	Nondepression Patients		Mild and Severe Depression Patients		Multivariate Logistic Analysis				Univariate Logistic Analysis			
	n	%	n	%	OR	95% CI		*p*-Value	OR	95% CI		*p*-Value
gender												
women	73	54.5%	49	74.2%	1.00							
men	61	45.5%	17	25.8%	0.52	0.25	1.08	0.079	0.42	0.22	0.79	0.008
comorbidities												
none	65	48.5%	3	4.5%	1.00							
at least 1	69	51.5%	63	95.5%	18.38	5.02	67.21	<0.001	19.78	5.92	66.13	<0.001
BMI												
<25	26	19.4%	4	6.1%	1.00							
≥25	108	80.6%	62	93.9%	0.97	0.25	3.81	0.970	3.73	1.24	11.19	0.019
HBA1C level												
<7%	46	34.3%	16	24.2%	1.00							
≥7% & <10%	75	56.0%	41	62.1%	1.87	0.87	3.99	0.107	1.57	0.79	3.12	0.195
≥10%	13	9.7%	9	13.6%	2.12	0.68	6.68	0.197	1.99	0.72	5.53	0.187

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
