# Peer review of "Prevalence of Depressive Symptoms in the Elderly Population Diagnosed with Type 2 Diabetes Mellitus"

_ijerph, 2020, doi:10.3390/ijerph17103553_

Round 1

Reviewer 1 Report

In the manuscript “Prevalence of depressive disorders in the elderly population diagnosed with type 2 diabetes mellitus” authors examined the prevalence of depressive symptoms in this segment of population. Results from the study show a statistically significant difference in the GDS for marital status, BMI, duration of diabetes, and the number of comorbidities. Patients with symptoms of mild and severe depressive disorders were found to have significantly higher BMI, longer duration of the disease, and more comorbidities.

The article is clear and well written and investigates a crucial topic in psychiatry. The abstract reflects the content of the article and it accurately describes what the author hoped to achieve. Results are presented in a logical sequence and table and figure are informative. In the Discussion section the authors summarized their findings properly and the speculations and extrapolations are reasonable.

Few suggestions below:

  1. Title and text: As you well described in the methods section, results from GDS are not equivalent to the diagnosis of depressive disorders, although they are diagnostically significat. For this reason, I think it would be better to say “depressive symptoms” instead of “depressive disorders” both in the title and in all the manuscript.
  2. Abstract: please add in the results section “Patients with symptoms of mild and severe depressive disorders were found to have significantly higher BMI, longer duration of the disease, and more comorbidities”. It is important to specify results for the different degrees of depression.
  3. Introduction and Discussion: please clarify if the aim of the study (lines 72-73 and 152-153) is “to evaluate the prevalence of depressive symptoms in the elderly 
population diagnosed with type 2 diabetes mellitus “ or to “to identify elderly people treated for type 2 diabetes mellitus who are at risk of developing depressive disorders”.
  4. Discussion: Please translate in English language sentences at lines 145-146.

Author Response

Dear Sir/Madam,

We would like to thank the Reviewers for taking the time to read our manuscript, and for their highly valuable comments and suggestions which significantly contributed to improving the quality of the paper.

We hope that the suggested revisions have been made correctly. All revisions in the text of the manuscript are marked in red and by means of the “Track changes” function, with reference provided to the number of the line where a change has been made.

Yours faithfully,

Authors

Comment 1

Title and text: As you well described in the methods section, results from GDS are not equivalent to the diagnosis of depressive disorders, although they are diagnostically significat. For this reason, I think it would be better to say “depressive symptoms” instead of “depressive disorders” both in the title and in all the manuscript.

Response

According to the Reviewer’s comments, the phrase “disorders” has been replaced by “symptoms” both in the title of the paper and throughout the entire manuscript (In lines 2,18,31,65,115,122,130,157,158,159,162,163,166,173,184,198,202,213,216,228).We are very grateful for this valuable suggestion.

Comment 2

Abstract: please add in the results section “Patients with symptoms of mild and severe depressive disorders were found to have significantly higher BMI, longer duration of the disease, and more comorbidities”. It is important to specify results for the different degrees of depression.

Response

According to the Reviewer’s suggestions, a sentence has been added in the Abstract (Results section) to present the results in a more detailed manner in relation to different stages of depression: “Patients with results indicative of symptoms of mild and severe depression were found to have higher BMI, longer disease duration, and a greater number of comorbidities”. (In lines 28-29). 

Comment 3

Introduction and Discussion: please clarify if the aim of the study (lines 72-73 and 152-153) is “to evaluate the prevalence of depressive symptoms in the elderly 
population diagnosed with type 2 diabetes mellitus “ or to “to identify elderly people treated for type 2 diabetes mellitus who are at risk of developing depressive disorders”.

Response

The aim of the paper has been unified throughout the manuscript. The aim of the study has been replaced by the following: Evaluation of the prevalence of depressive symptoms in the elderly population diagnosed with type 2 diabetes mellitus. (In Lines 18-19,74-75,154-155).

Comment 4

Discussion: Please translate in English language sentences at lines 145-146.

Response

According to the Reviewer’s suggestions, the following sentence has been translated into English: “Depression is one of the leading causes of a reduction in disability-adjusted life years globally” (Line 147).

Other comments:

Response

According to the Reviewer’s suggestion, in order to improve the transparency of reporting study results, the following revisions have been made:

  • The title of Table 1 has been changed to: Correlations between sociodemographic variables and GDS scores. (Line 117).
  • The title of Table 2 has been changed to: Correlations between clinical variables and GDS scores. (Line 133).
  • The title of Table 3 has been changed to: Correlation between depression symptoms and the duration of diabetes (years). (Line 134).
  • The title of Table 4 has been changed to: Correlations between comorbidities and GDS scores. (line 135).
  • Table 5 has it is with the following revisions made in the header:
    • Healthy subjects to Non-depression patients (Line 139,142).
    • Patients to Mild and severe depression patients (Line 139,142).

Reviewer 2 Report

In the article entitled Prevalence of depressive disorders in the 2 elderly population diagnosed with type 2 diabetes mellitus, Dziedzic and coworkers provide information considering the role of processes accompanied with aging on the prevalence of depressive disorders. This epidemiological study is a very well-written and well organized, with adequate information on the subject. The data presented help to understand various variables and their relation to this common mood disorder. I have no major comments or concerns for this article.

Since I am not native speaker so I did not comment language and style. Also, as you asked for more specific comments I can confirm that:

  • the topic is actual
  • epidemiological data, although obtained on 200 subjects, are sufficient for analysis
  • since this is not the type of investigation that requires fundamental disclosure, I think that it still may provide useful information for clinicians
  • the whole concept is not pretentious, so the data presented in the manuscript fulfill its intention
  • the whole manuscript is well organized with appropriate methodology
  • the references are relevant and updated

Author Response

Dear Sir/Madam,

We would like to thank the Reviewers for taking the time to read our manuscript, and for their highly valuable comments and suggestions which significantly contributed to improving the quality of the paper.

We hope that the suggested revisions have been made correctly. In the manuscript, they are marked in red and by means of the “Track changes” function, with reference provided to the number of the line where a change has been made.

Yours faithfully,

Authors

Reviewer 3 Report

In general, the authors examined the prevalence of depressive disorders among 200 older adults diagnosed with type 2 diabetes mellitus in Poland.

The research questions are well defined and the results are interpreted properly. The manuscript is well written. The data and analyses performed are appropriate.

The study methodology is clearly described. The conclusions are well supported by the study findings and results. English language is appropriate and understandable.

Please correct the typo in the abstract - results. "A statistically significant difference in the GDS (p<00.1) was shown for marital status,...

Author Response

Dear Sir/Madam,

We would like to thank the Reviewers for taking the time to read our manuscript, and for their highly valuable comments and suggestions which significantly contributed to improving the quality of the paper.

We hope that the suggested revisions have been made correctly. In the manuscript, they are marked in red and by means of the “Track changes” function, with reference provided to the number of the line where a change has been made.

Yours faithfully,

Authors

Comment 1

Please correct the typo in the abstract - results. "A statistically significant difference in the GDS (p<00.1) was shown for marital status,...

Response

According to the Reviewer’s suggestion a spelling error has been corrected in the Abstract – Results section p<0.01 (line 27).
